

# Effect of vitamin C injections on exercise muscular performance and biochemical parameters in *Trichinella spiralis*-infected mice

Hadeer Abd El-hak Rashed[1], Bander Albogami[2], Abdulsalam A. M. Alkhaldi[3], Najlaa Y. Abuzinadah[4], Samah S. Abuzahrah[4], Fawziah A. Al-Salmi[2], Eman Fayad[5], Rewan Mohamed Fouad[1], Manar Elsayed Fikry[1], Abd-Allah Ahmed ElSaey[1] and Ali Hussein Abu Almaaty[1]

[1] Zoology Department, Faculty of Science, Port Said University, Port Said, Egypt
[2] Department of Biology, College of Sciences, Taif University, Taif, Saudi Arabia
[3] Department of Biological Sciences, College of Science, Jouf University, Sakaka, Saudi Arabia
[4] Department of Biological Sciences, College of Science, University of Jeddah, Jeddah, Saudi Arabia
[5] Department of Biotechnology, College of Sciences, Taif University, Taif, Saudi Arabia

## ABSTRACT

**Background**. *Trichinella spiralis* is a worldwide intestinal nematode that can parasitize the striated muscles of its hosts at the larval stage. This study aims to evaluate potential of vitamin C for treating trichinellosis-related pathological problems in the infected muscles of mice.

**Materials and Methods**. Thirty CD1 male Albino mice were divided into three groups (10 mice per group). Negative and positive control groups (0.9% NaCl) and the infected vitamin C group (10 mg/kg body weight). Two weeks post-infection, each group was intraperitoneally injected daily for two weeks with Vitamin C or saline. The performance of the muscles was assessed both before and after the treatment. After dissection, constant parts of striated muscles were removed for further assays. The scoring of the histological changes of infected muscles was carried out. In addition to muscle malondialdehyde levels, superoxide dismutase and catalase activities were measured for the oxidative and antioxidant states. Creatine kinase and aspartate aminotransferase were also measured in tissues to reflect the degree of muscular damage.

**Results**. Vitamin C enhances the weakness of the muscular performance resulting from the infection. Vitamin C was able to repair some of the histological lesions that resulted from the infection. Trichinellosis caused severe changes in the biochemical markers in positive control animals. Muscle damage biomarkers and, besides, oxidative and antioxidant conditions were greatly ameliorated in infected vitamin C animals. Summing up, vitamin C can be used as a complementary drug due to its efficiency in improving pathogenesis following a trichinellosis infection. The supplement also must be tested in the intestinal stage of infection after showing promising results in the muscular stage.

Corresponding authors
Hadeer Abd El-hak Rashed,
hader_abdelhak@sci.psu.edu.eg
Eman Fayad, e.esmail@tu.edu.sa

## INTRODUCTION

Nematodes of the genus *Trichinella* are among the most prevalent zoonotic parasites found in the muscles of several terrestrial vertebrate species from all continents except Antarctica (*Pozio, 2007*). In the human host, trichinosis has two stages: intestinal and muscular (*Bruschi & Dupouy-Camet, 2014*).

*Trichinella spiralis* developmental stages can be found in one host. The infection starts with a brief intestinal phase, where the infective mature larvae transform into adult worms. The adult parasites, in turn, discharge the migrating young larvae (*Ilic, Gruden-Movsesijan & Sofronic-Milosavljevic, 2012*). In the skeletal muscles, the *T. spiralis* larvae develop into fully infective muscle larvae and modify the muscle cells into a new architecture called nurse cells. The parasite uses this technique to survive in the host for several months to years (*Ilic, Gruden-Movsesijan & Sofronic-Milosavljevic, 2012*; *Sun et al., 2019*).

*T. spiralis* larvae cause an appropriate inflammatory response once they have penetrated the skeletal muscle tissue, which is responsible for auto-muscular atrophy or myositis. The larva itself and the inflammatory cells' presence generate many free radicals, including reactive oxygen species, contributing to host tissue damage (*Chiumiento & Bruschi, 2009*).

The two main anthelmintic medications used to treat trichinellosis are albendazole and mebendazole (*Gottstein, Pozio & Nöckler, 2009*). However, according to *Caner et al. (2008)*, they have low bioavailability, a high level of resistance, and limited efficacy against the encapsulation of larvae in the muscular phase.

Ascorbic acid, also known as vitamin C, is one of the main and best-known micronutrients essential for proper of human body (*Berdanier, 2022*). Ascorbic acid was recorded to have antioxidant and anti-inflammatory properties (*Gęgotek & Skrzydlewska, 2022*), and it is also used as a complementary supplement to anti-chagasic drugs to diminish their cytotoxicity (*Puente et al., 2018*). Ascorbic acid was reported to have some impacts on inflammatory mediators and muscular damage during the muscle healing regeneration process in rats (*Işin et al., 2023*).

Consequently, this study aimed to evaluate the ability of vitamin C to reduce the effects of different *T. spiralis* larvae in the striated muscles and improve muscle performance. The effectiveness evaluation was designed using parasitological, histological, and biochemical marker assays.

## MATERIALS AND METHODS

### Ethics statement

All studies were approved by the research animal care ethical committee of the Faculty of Science, Suez Canal University with approval reference number of REC249/2023.

**Table 1 Experimental design.**

| Group (N = 10/group) | Dose | Mode of injection |
|---|---|---|
| Uninfected + saline (-Ve control) | 0.1 mL/mouse weight of 0.9% NaCl | |
| Infected + saline (+Ve control) | 0.1 mL/mouse weight of 0.9% NaCl | Intraperitoneally |
| Infected + vitamin C | 0.1/ mL mouse weight of vitamin C (10 mg/kg bodyweight). The dose was according to *Binfaré et al. (2009)*. | |

Notes.
N, number of mice.

## Chemicals

Vitamin C effervescent tablets were purchased from Chemical Industries Development Company (CID), Giza, Egypt. Creatine kinase (CK) and aspartate aminotransferase (AST) kits were purchased from Clinicchem Company, Budapest, Hungary. Malondialdehyde (MDA), catalase (CAT), and superoxide dismutase (SoD) kits were brought from Cell Biolabs Inc., San Diego, CA, USA.

## Preparation of vitamin C

The orally administrated effervescent tablets were prepared for the intraperitoneal injection dissolved in 0.9% NaCl. The solution was then filtrated using 0.22 $\mu$m filter paper to remove any undissolved particles or impurities.

## Experimental animals and parasites

Thirty CD1 male white albino mice (15–25 g) were brought from Theodor Bilharz Research Institute in Giza, Egypt. After that, they were housed in the Faculty of Science's Animal House in the Zoology Department of Port-Said University in plastic cages with 10 mice each. They were kept in regular settings with a 12-hour light/dark cycle and room temperature (20–25 °C). They were fed a regular diet consisting of 10% ground yellow corn, 20% milk, 10% vegetables, and 50% ground barley. Throughout the trial period, food and water were freely available for consumption. Before experimentation, animals were given a 5-day rest period to allow for adaptation. *T. spiralis* isolate was obtained from infected albino mice from Theodor Bilharz Research Institute, Giza, Egypt.

## Experimental design

The mice were divided into three groups: (10 mice per group) as shown in Table 1. Twenty mice received 200 *T. spiralis* larvae/mouse orally (*Dunn & Wright, 1985*), while 10 mice were kept as negative control parasitic-free. Fifteen days post-infection, all the mice began to receive their daily injections (vitamin C or saline) for two weeks (Table 1).

## Changes in the body weight

The animals of the different groups were weighed every week during the experiment. The percent of the change in the body weights in the last week was calculated in comparison

with the beginning of the experiment in the first week as follows:

$$\left(\frac{Lw - Fw}{Lw}\right) \times 100$$

where, **Lw** → The average of the body weights in each group in the last week.

    **Fw** → The average of the body weights in each group in the first week.

## Measuring muscle performance

The performance was detected according to two models (hanging and weights). The two assays targeting shoulders, back, chest and fore and hind limbs muscles. They were applied during the pre-treatment and post-treatment periods. The pre-treatment stage represents the resting stage before the spread of the larvae in the motor muscles. Mice performed the tests on the last day before either infection or dissection, representing the pretreatment and post-treatment periods, respectively.

## Hanging test

The wire hang test aims to assess rodent model motor function and impairment. The mice were permitted to grasp a 2-mm-diameter wire made of metal that was held horizontally at a distance above a thick layer of soft bedding using their two paws (Fig. 1). The time until the mice dislodged themselves from the wire was noted. The mice were given a minute to recover after each fall. There were three trials total in each session, and the scores averaged (*Dorchies et al., 2013*).

## Weights test (muscle strength)
### *Principle*
The apparatus was made of a stainless-steel wire that was tangled in the form of a small ball weighing 2 g. This ball was attached to a steel chain link with a constant weight of 1.3 g. The number of links ranges from one to seven. Thus, the total weight of the first apparatus was 3.3 g, while the last one weighs 11.1 g, as shown in Fig. 2A. This technique was according to *Deacon (2013)*, with changes in the used weights.

### *Procedure*
Hold a mouse by the center or base of the tail and drop it to allow the animal to grasp the first weight. Three-second holds were standard; if the mouse can lift the weight for three seconds, try with the next heavier object (Fig. 2B). Three tries were provided to each mouse to hold the weight for three seconds. The final score was calculated according to the following equation:

$$\mathbf{F^s} = (\mathbf{L} \times 3) + D_s$$

where,

$\mathbf{F^s}$ → Finalscore

$\mathbf{L}$ → The number of links in the heaviest chain held for the full 3 seconds.

$\mathbf{D_s}$ → The duration, in seconds, during which the mouse was unable to maintain the last weight for three seconds.

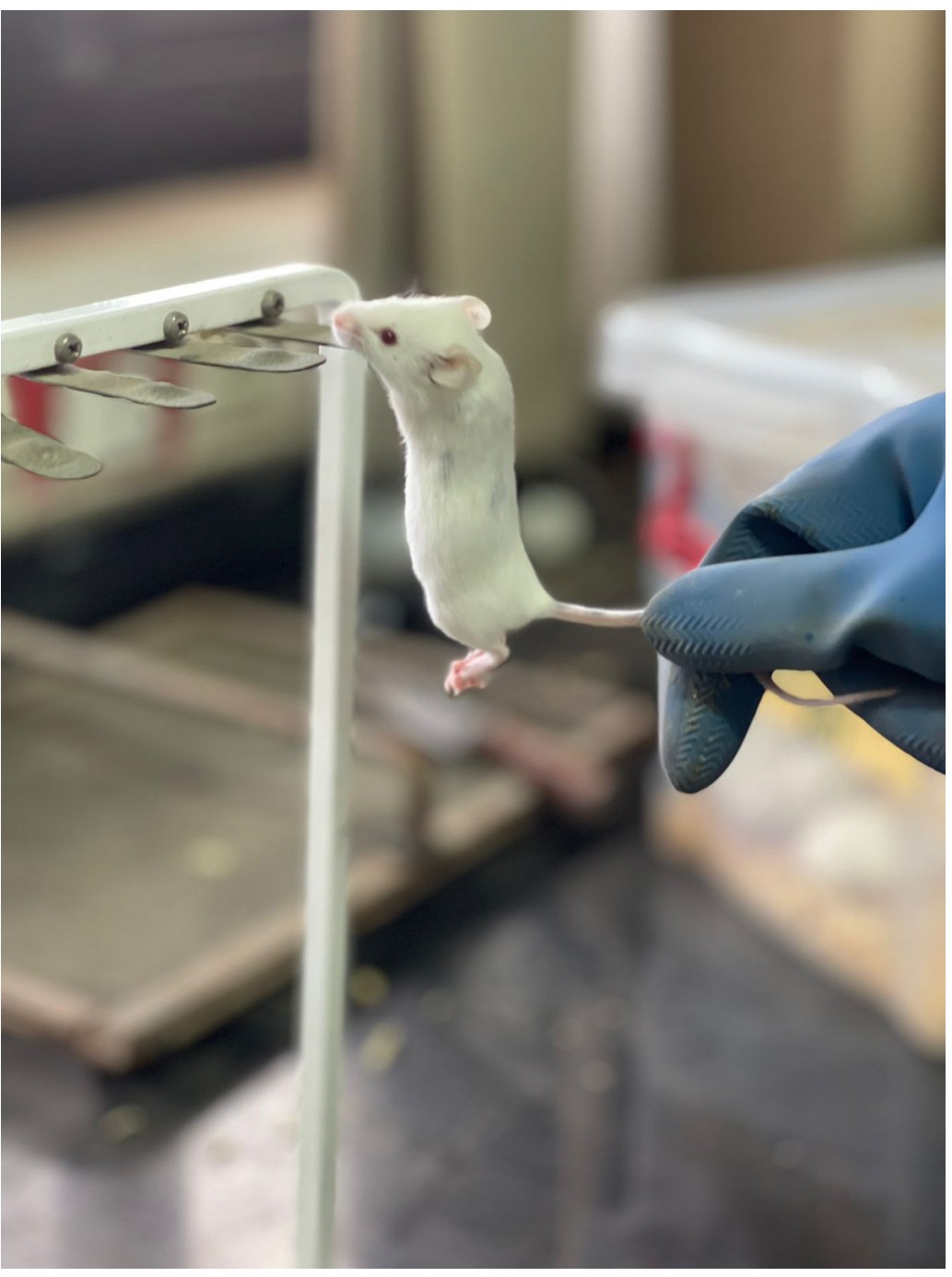

**Figure 1** Hanging test.

## Dissection and separation of muscles

The animals were given  isoflurane anesthesia after thirty days post-infection, which allowed for a rapid and painless loss of consciousness (*Miller et al., 2015*). Euthanasia was carried out by carotid exsanguination. Following their dissection, the mice's bodies were

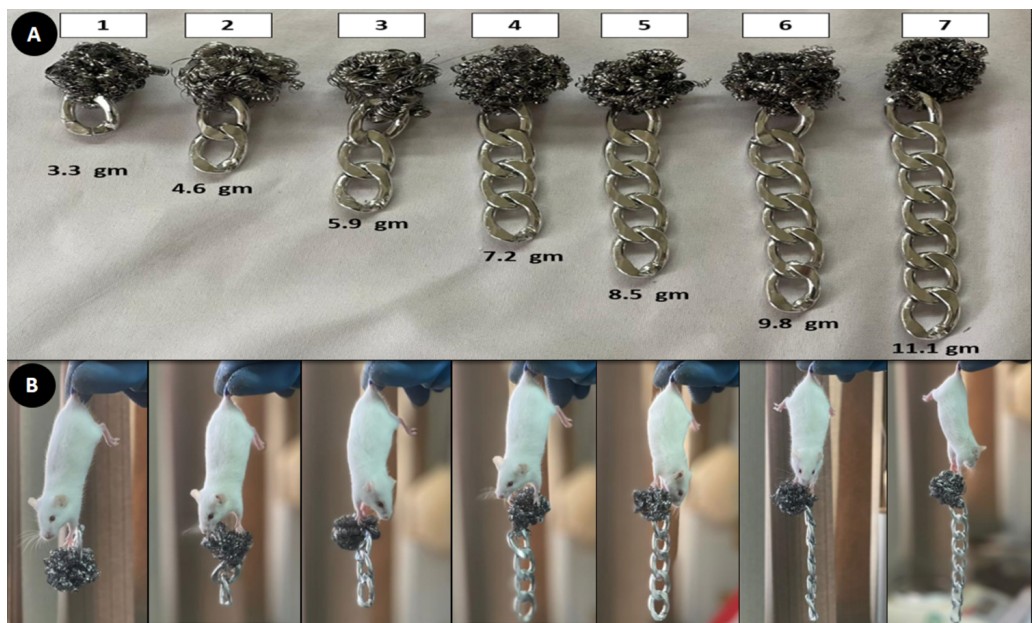

**Figure 2 Muscle strength assay.** (A) Seven weights (3.3 gm–11.1 gm) (B) Mice holding the different weights.

cleaned with tap water to get rid of any hairs. The same parts of muscles in the different groups were removed for further parasitological, histological, and biochemical assays.

## Plate compression technique

This technique was used to check the spread of the encysted larvae (Fig. 3) in the striated muscles of the infected animals. For this assay, a constant weight (0.1 g) and portion of different muscles from each infected group was used. Each sample was placed between two glass slides and compressed; it was observed under the light microscope with the 10 × and 40 × lenses (*Gamble, 1996*).

## Histopathological studies

As soon as the animals were dissected, samples of striated muscles were taken out and fixed in 10% buffered formalin. After the fixation process, samples were embedded in paraffin and cut into blocks with a thickness of 4 μm before being stained with hematoxylin and eosin. Using a light microscope, the mounted samples were examined to be scored. The structural alterations and histological problems in the tissue sections of the various groups were compared using scoring. Statistical grading of the lesions (fibrosis, cellular infiltration, nurse cell, vacuolation, and muscular destruction) was done according to *Landmann et al. (2021)*, with minor modifications as the lesions were classified as follows:

0 → **Null lesion**

1 → **Mild lesion**

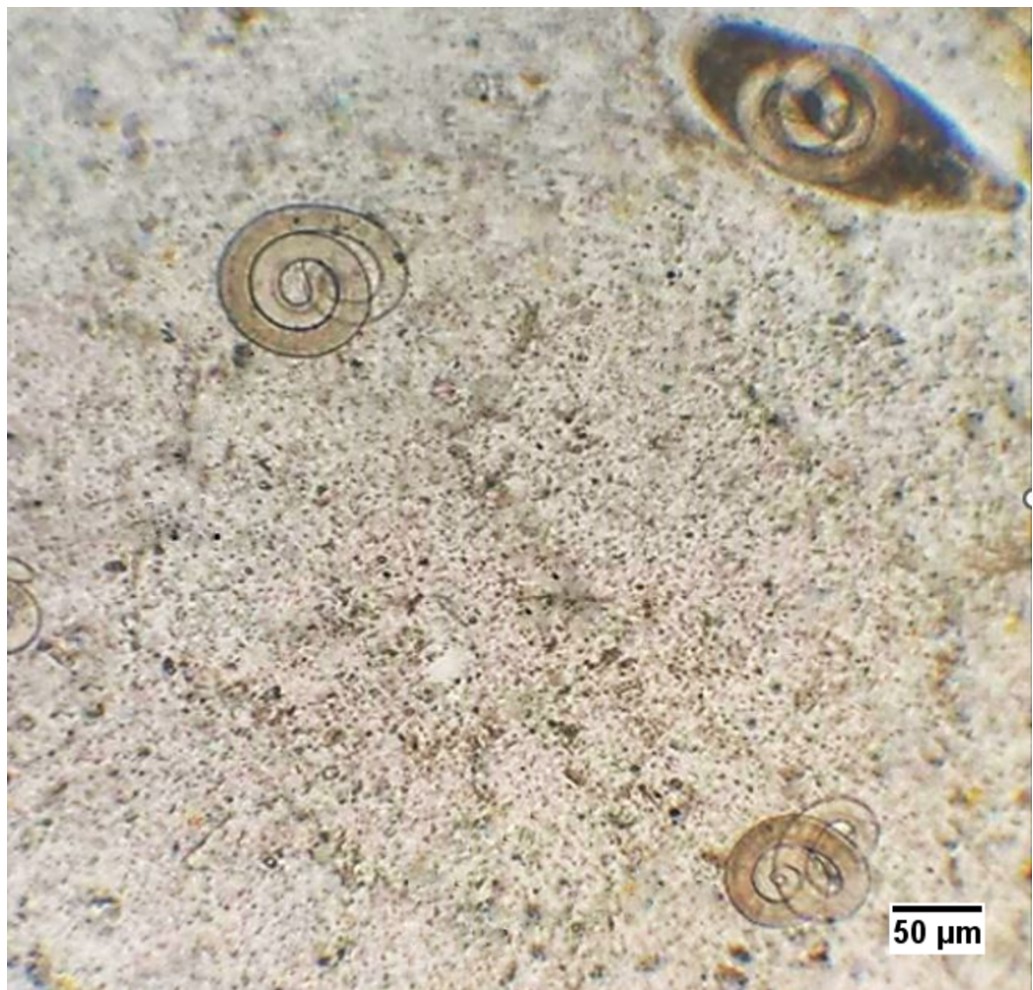

**Figure 3** *T. spiralis* larvae in the infected mice.

2 → **Moderate lesion**

3 → **Severe lesion**

In addition to the histological abnormalities, the larvae were categorized statistically based on the degree of integrity of their internal structures. The larvae without any changes in their structures were described as intact. Segmented into two pieces or more, these larvae were defined as partially intact or deformed. In The last type, the larvae inside their capsule were completely replaced with vacuoles.

## Muscle-damage biochemical biomarkers

The activities of Ck and AST in the muscles were performed according to the methodology of *Hess et al. (1964)* and *Bergmeyer, Herder & Ref (1986)*, respectively.

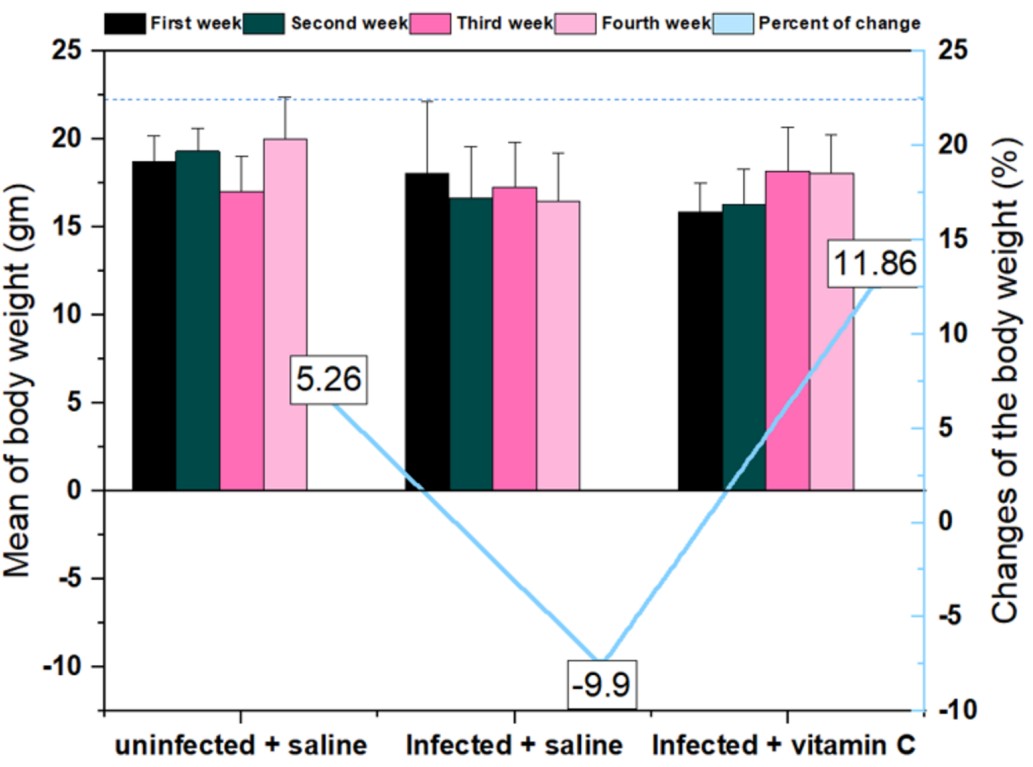

**Figure 4** Mean and changes in the body weights of mice during the experimental period # significant difference from the infected + saline group ($N = 10$).

## Oxidative stress and antioxidant biochemical parameters in muscles

MDA Level of muscle tissue was assessed according to *Ohkawa, Ohis & Yagi (1979)*. While SOD and CAT activities were measured according to *Oyanagui (1984)* and *Johansson & Borg (1988)*, respectively.

## Statistical analysis

The data was displayed as mean and standard deviation (mean ± SD). The statistical programme Sigma Stat (SPSS), version 20, was used for data analysis. Analysis of variance, one-way ANOVA, was used to examine the effects of various regimens, and the Tukey *post-hoc* test was then performed. A statistically significant value was defined as $P \leq 0.05$.

## RESULTS

According to the results, trichinellosis caused a reduction in the average body weight over time in the infected animals treated with saline. The bodyweight of this group showed a reduction (in the average from the first week ($18.1 \pm 4.07$) to the fourth week ($16.5 \pm 2.70$). In contrast, the negative control and infected vitamin C groups displayed a considerable increase in body weight of 5.26% and 11.86%, respectively, at the end of the experiment compared with the first week (Fig. 4).

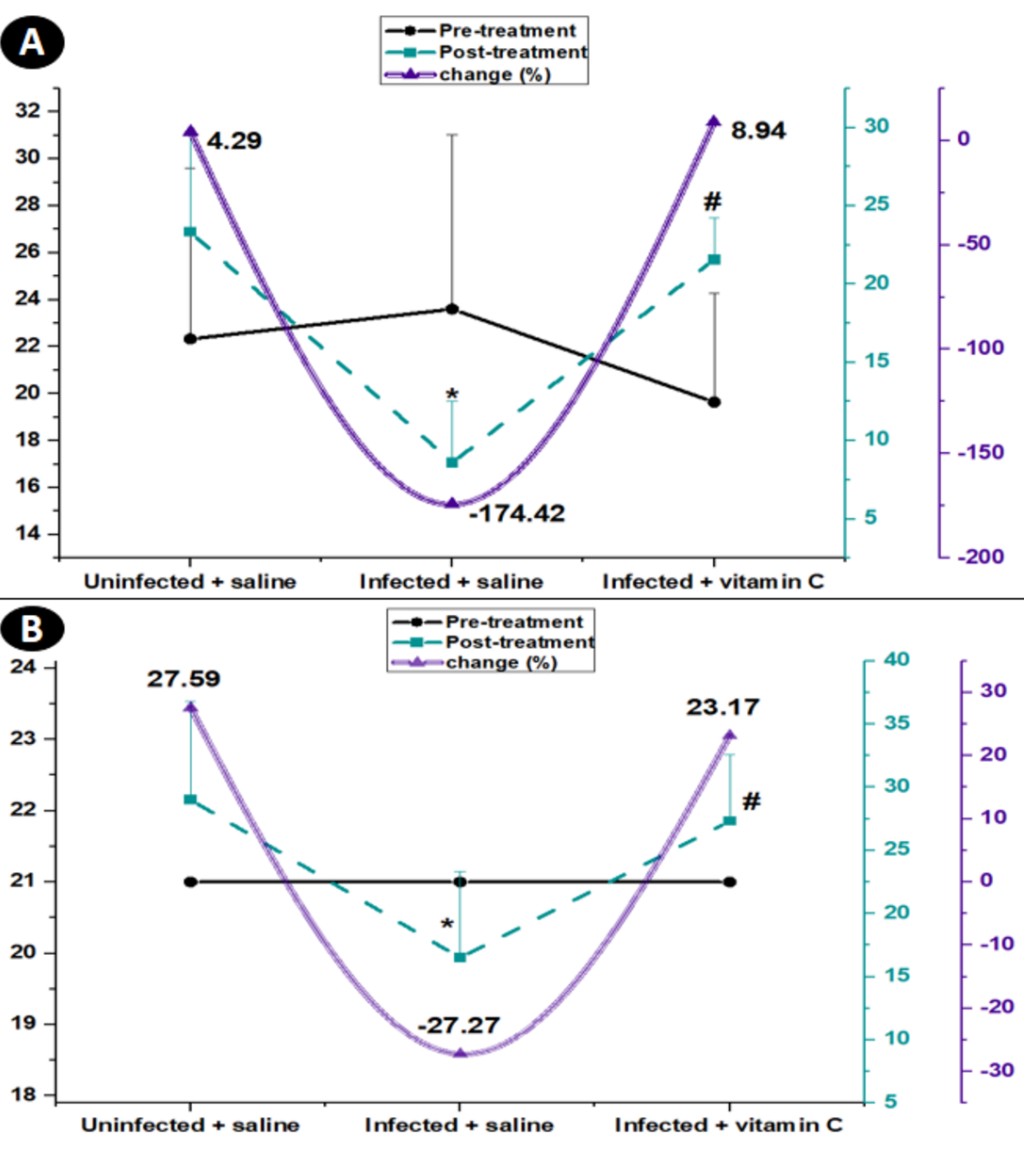

**Figure 5** **Average of muscle performance in the different groups with the rate of change from pre-treatment to post-treatment.** (A) Time latency to fall (Wire hanging). (B) Muscle strength (weights test). * Significant difference from the un-infected group, # significant difference from the infected + saline group ($N = 10$).

In the pre-treatment stage, the different groups showed no significant variations in the two assays of the muscular performance efficiency (weighted test and hanging test) (Fig. 5).

In the post-treatment stage, the average latency to fall in the hanging test increased by 4.29 in the normal animal injected with saline. The positive control mice recorded a decline in latency to $8.60 \pm 3.29$. The infected group injected with vitamin C displayed a post-treatment raise of 8.94% with a highly significant change ($P < 0.001$) from the infected mice treated with saline (Fig. 5A).

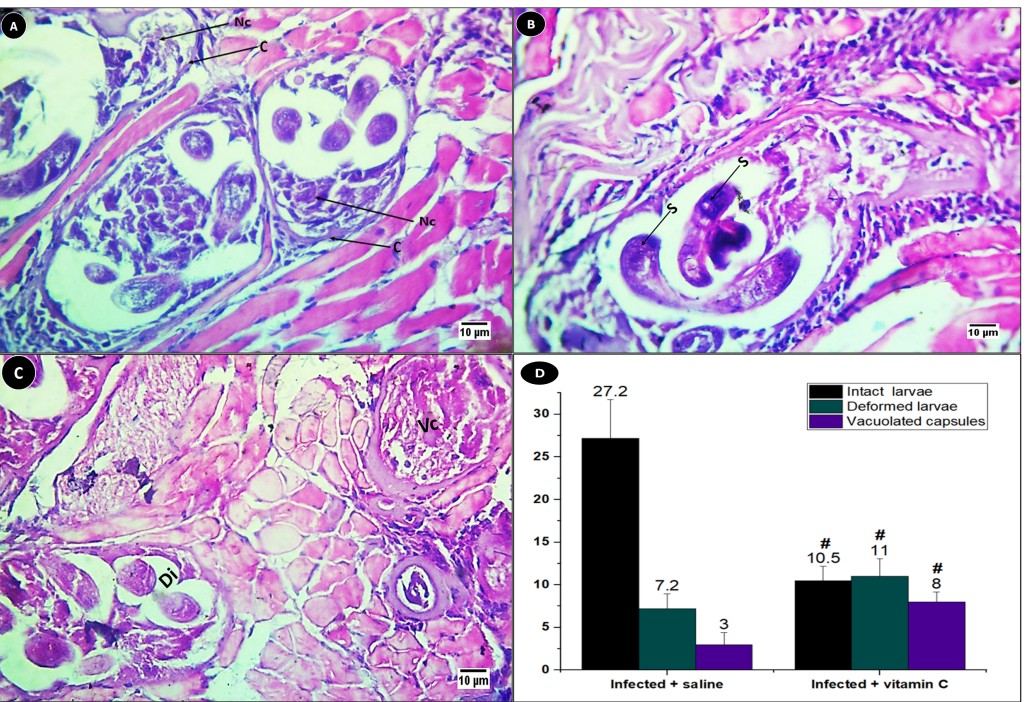

**Figure 6 Different encysted larvae types in the muscles of infected mice.** (A) Structure of the encysted larvae. (B–C) Histological sections showing the structures of each larvae type. (D) Mean of different encysted larvae types. Nc, nurse cell; C, Capsule; S, stichosomes of intact larvae; Dl, deformed larvae; Vc, a vacuolated capsule. # significant difference from the infected + saline group ($N = 10$).

Regarding the post-treatment in the weights test, the musculature power of the normal control mice increased by 27.59%. Over time and with increasing the spread of infection, the musculature strength in the infected mice injected with saline dropped by $-27.27\%$ in the post-treatment period. The infected animals injected with vitamin C displayed a post-treatment rise of 23.17% with a highly significant change ($27.33 \pm 5.23$, $P < 0.01$) from the infected mice treated with saline only (Fig. 5B).

In the infected groups, *T. spiralis* larvae were submerged intracellularly within the muscle fiber tissues. The encysted larvae can be differentiated by the presence of a nurse cell that was surrounded by the developing collagen capsule, as shown in Fig. 6A.

The larvae were encysted in three forms: intact, degenerated, and completely replaced with vacuoles. The intact larvae have a curled terminal end with stichosomes that were darkly stained (Fig. 6B). While the second type of larvae disintegrated into small pieces inside the capsule, the third kind the larvae were replaced with vacuoles (Fig. 6C).

There was a significant difference ($P < 0.001$) in the average of three forms of larvae between the infected animals treated with saline and those treated with vitamin C. The intact larvae recorded the highest average ($27.2 \pm 4.52$) in the infected mice injected with saline only. In the group treated with vitamin C, the highest average ($11 \pm 2.05$) was recorded for the deformed larvae (Fig. 6D).

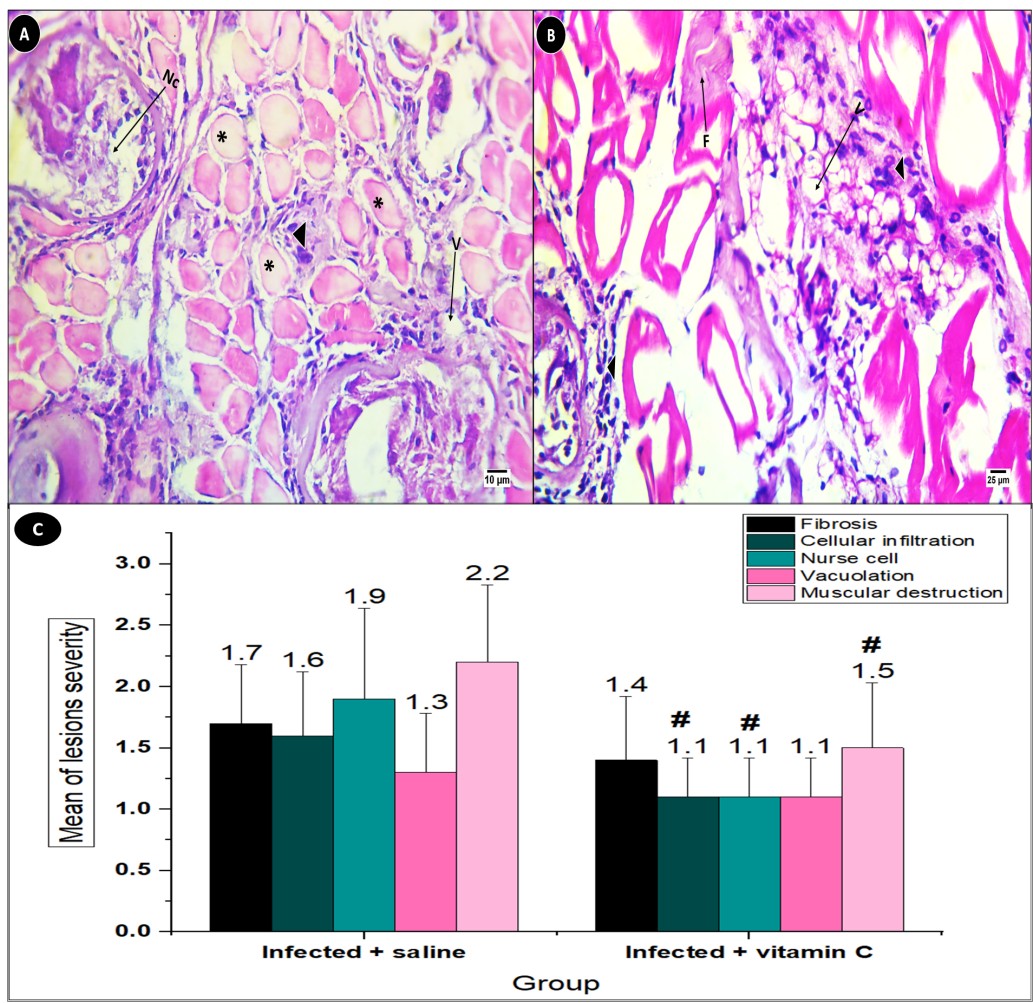

**Figure 7  Skeletal muscle pathological lesions in *T. spiralis*-infected groups.** (A) Mice treated with saline. (B) Mice treated with vitamin C. (C) Mean of histopathological lesions severity. F, fibrosis; V, vacuolation black arrowhead: cellular infiltration, asterisk: muscle cell destruction. # significant difference from the infected + saline group ($N = 10$).

The histopathological lesions of fibrosis, cellular infiltration, nurse cells, vacuolation, and muscle damage were compared to the infected mice receiving vitamin C injections with the untreated group (Fig. 7). The increase in fibers indicated cellular infiltration, particularly in the vicinity of the growing nurse cells. (Figs. 7A & 7B) illustrates the numerous vacuolations that were seen between the inflammatory regions. General indicators of muscle cell damage included swelling and a hyaline appearance (Fig. 7A).

The severity of the lesions varied across the two infected groups, with the saline group suffering lesions with greater severity than the vitamin C group. In comparison with the positive control group ($2.20 \pm 0.63$), the muscle damage in the vitamin C-treated group showed a great improvement ($1.50 \pm 0.53$, $P < 0.01$). On the other hand, vacuolation showed the least improvement in the groups treated with vitamin C as compared with

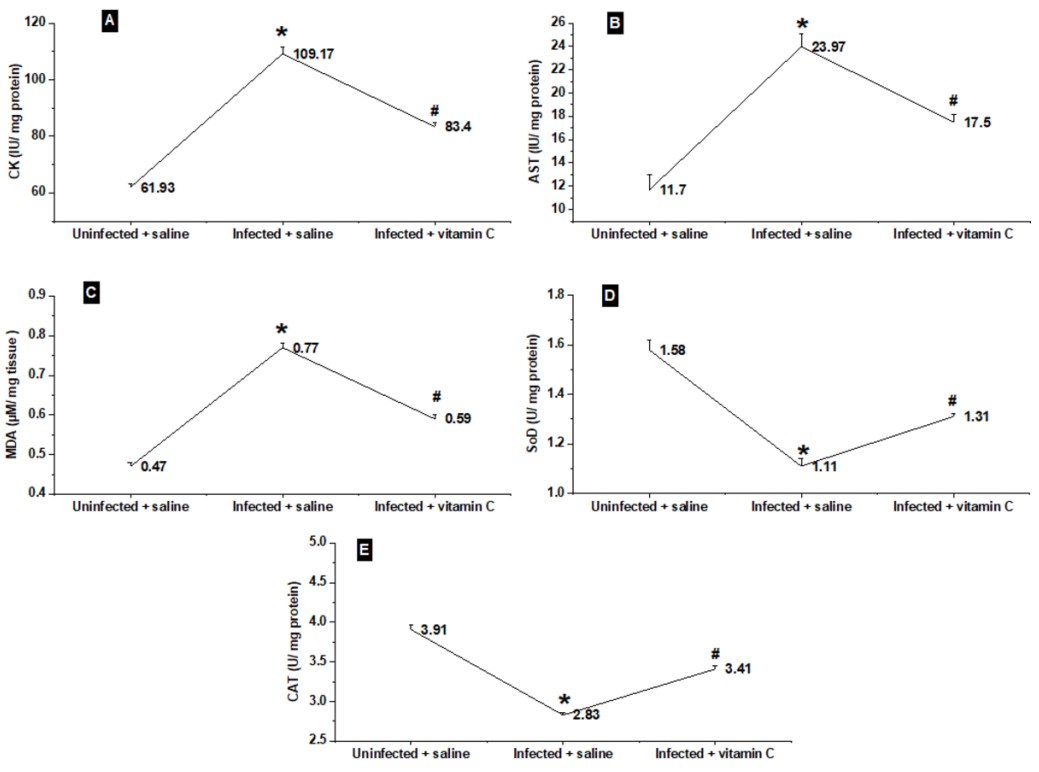

**Figure 8 Physiological biomarkers changes in the different groups.** (A–B) Muscle injury biomarkers. (C) Oxidative stress biomarker. (D–E) Antioxidants biomarkers. *significant difference from the uninfected group, # significant difference from the infected + saline group ($N = 10$).

saline-positive control animals ($1.10 \pm 0.32$ and $1.30 \pm 0.48$, respectively), as illustrated in Fig. 7C.

The activities of CK and AST in the striated muscles of the different groups are shown in (Figs. 8A & 8B). The highest average of CK and AST activity was recorded in the infected animals injected with saline ($109.17 \pm 2.16$ and $23.97 \pm 1.10$, respectively). The infected-vitamin C group showed a significant diminishment ($P < 0.001$) in CK and AST activities in comparison with the infected-saline mice.

The changes in the average MDA levels of mice tissues in different groups are shown in Fig. 8C. The infected animals treated with saline only showed a significant elevation ($P < 0.001$) in MDA levels as compared with the saline control group. On the other hand, the infected mice treated with vitamin C exhibited a significant decrease in MDA level to $0.59 \pm 0.01$ as compared with the positive control group.

The antioxidant parameters, represented by SoD and CAT, in the muscles of the three groups are shown in Figs. 8D & 8E. In a different manner from CAT, AST, and MDA results, the infected mice injected with saline displayed the lowest averages of both SoD and CAT. The infected vitamin C group exhibited a significant increase ($P < 0.001$) in SoD and CAT activities ($1.31 \pm 0.03$, and $3.41 \pm 0.04$, respectively) in comparison with the infected-saline mice.

## DISCUSSION

This work presents one of the few studies dealing with the immunomodulatory effect of vitamins on the various types of parasitic helminths, as most studies focus on natural plant extracts (*Abu Almaaty et al., 2021*; *El-Kady et al., 2022*; *Gogoi, Soren & Yadav, 2022*).

Trichinellosis decreased the body weight by $-9.9\%$ over time in infected mice injected with saline only. This agreed with *Li et al. (2023)* who found that at 30 days post-infection, the body weight of the infected group was significantly decreased in comparison with the control animals. The animals treated with vitamin C supplements were at an equal distance from the two groups (negative and positive control).

In both tests of muscle performance, the infected-saline group showed a drop in efficiency between the statistics after 7 days-pi (pre-treatment) and (post-treatment). This effect of trichinellosis agreed with *Park et al. (2018)* who found that 3 to 6 weeks pi, *T. spiralis*-infected mice's muscular strength significantly declined as compared with control mice. Vitamin C infected-treated animals showed a significant rise in the post-treatment animals in comparison with infected-untreated mice. This could be the main role of vitamin C in preventing muscle weakening and enhancing physical performance (*Takisawa et al., 2019*).

Mice injected with 10 mg/body weight (ip) of vitamin C showed an increase in the levels of IL-4 and IL-5. Both IL-4 and IL-5 are essential for eosinophil production (*Sindi, 2023*), which in turn plays a key role in the elimination of helminth larval stages (*Huang & Appleton, 2016*). Following this immunological role of vitamin C, the deformed larvae in the histological sections of the present study significantly increased in the animals treated with vitamin C compared to those injected with saline only ($11 \pm 2.05$, and $7.20 \pm 1.75$, respectively). This hypothesis agreed with *El-Dardiry et al. (2021)*, who reported that immune-competent *T. spiralis*-sensitized mice showed an increase in the degenerated larvae, which were finally replaced by hyaline matter.

The larval invasion of myocytes was followed by the development of molecular and metabolic units to provide the parasite with nutrients and protect them as much as possible; thus, the muscle fibres were replaced into a complicated structure known as nurse cells. These newly formed nurse cells attract an inflammatory response represented by mononucleated cells surrounding these areas (*Wu et al., 2008*). This illustrates the significant coordinated reduction in both the number of nurse cells and inflammatory infiltration in the mice treated with vitamin C in comparison with the positive control group. Swelling and a hyaline appearance of myocytes were the signs of muscle cell destruction in this work. This description agreed with that recorded by *Elguindy et al. (2019)*. The reduction of nurse cells and the other histological lesions in the vitamin C-infected group caused a noticeable improvement in muscle cell atrophy.

Increases in CK and AST are indicative of muscle cell damage or recently diagnosed or active myonecrosis (*Ross & Dyson, 2010*). So, in this study, both parameters were assessed to reflect the damage to the infected muscles. As expected, the infected-saline groups showed significant elevations in the activities of both enzymes. The infected mice treated with vitamin C showed a significant decrease in the activity of these enzymes. The

later point coordinated with the previously mentioned improvements in the muscular conditions at either the parasitological or histological levels.

In contrast to the mice in the negative control group, the muscles of the infected-control animals had significantly higher levels of MDA but lower enzymatic antioxidants activities. In agreement with our results, *Hamed et al. (2022)* reported that level of MDA was elevated during the infection's muscular phase.

Muscles of *T. spiralis*-positive control mice showed a significant decrease in their total antioxidant capacity (*Elgendy et al., 2020*). Vitamin C can be defined as a hydrophilic antioxidant and can act directly by scavenging lipid hydroperoxide, superoxide, and hydroxyl radicals (*Powers & Jackson, 2008*). This explains the effectiveness of vitamin C in the current investigation by significantly improving the antioxidant activities in the muscles of the infected mice when compared with the animals used as positive controls.

## CONCLUSION

In conclusion, it has been demonstrated that vitamin C improved muscle performance after treatment in *T. spiralis-* infected mice. The above-mentioned finding was supported by the histological sections, which showed significant ameliorative differences between vitamin C-infected animals and positive control mice. According to these optimistic findings, the vitamin C-infected group of mice had significantly lower activity of the biomarkers (CK and AST) indicative of muscle injury than the +ve control animals. In addition, the antioxidant and oxidative stress conditions were improved. There is a need to test vitamin C in combination with albendazole supplement in the intestinal phase of trichinellosis. Besides, we need to isolate nonintact larvae and infect a new generation of animals to measure the ability of these larvae to cause trichinellosis.

### Funding
This research was funded by Taif University, Taif, Saudi Arabia (Tu-Dspp-2024-155). The funders had no role in study design, data collection and analysis, decision to publish, or preparation of the manuscript.

### Grant Disclosures
The following grant information was disclosed by the authors:
Taif University, Taif, Saudi Arabia: Tu-Dspp-2024-155.

### Competing Interests
The authors declare there are no competing interests.

### Author Contributions
- Hadeer Abd El-hak Rashed conceived and designed the experiments, performed the experiments, analyzed the data, prepared figures and/or tables, authored or reviewed drafts of the article, main power for research, and approved the final draft.

- Bander Albogami conceived and designed the experiments, performed the experiments, analyzed the data, prepared figures and/or tables, authored or reviewed drafts of the article, and approved the final draft.
- Abdulsalam A.M. Alkhaldi conceived and designed the experiments, performed the experiments, analyzed the data, prepared figures and/or tables, authored or reviewed drafts of the article, and approved the final draft.
- Najlaa Y. Abuzinadah conceived and designed the experiments, performed the experiments, analyzed the data, prepared figures and/or tables, authored or reviewed drafts of the article, and approved the final draft.
- Samah S. Abuzahrah conceived and designed the experiments, performed the experiments, analyzed the data, prepared figures and/or tables, authored or reviewed drafts of the article, and approved the final draft.
- Fawziah A. Al-Salmi conceived and designed the experiments, performed the experiments, analyzed the data, prepared figures and/or tables, authored or reviewed drafts of the article, and approved the final draft.
- Eman Fayad conceived and designed the experiments, performed the experiments, analyzed the data, prepared figures and/or tables, authored or reviewed drafts of the article, and approved the final draft.
- Rewan Mohamed Fouad conceived and designed the experiments, performed the experiments, analyzed the data, prepared figures and/or tables, authored or reviewed drafts of the article, and approved the final draft.
- Manar Elsayed Fikry conceived and designed the experiments, performed the experiments, analyzed the data, prepared figures and/or tables, authored or reviewed drafts of the article, and approved the final draft.
- Abd-Allah Ahmed ElSaey conceived and designed the experiments, performed the experiments, analyzed the data, prepared figures and/or tables, authored or reviewed drafts of the article, and approved the final draft.
- Ali Hussein Abu Almaaty conceived and designed the experiments, performed the experiments, analyzed the data, prepared figures and/or tables, authored or reviewed drafts of the article, and approved the final draft.

### Animal Ethics

The following information was supplied relating to ethical approvals (i.e., approving body and any reference numbers):

Faculty Science, Suez Canal University under protocol REC249/2023.

### Ethics

The following information was supplied relating to ethical approvals (i.e., approving body and any reference numbers):

All studies were approved by the research animal care ethical committee of the Faculty Science, Suez Canal University under protocol REC249/2023.

### Data Availability

The raw data is available in the Supplemental Files.

## Supplemental Information

Supplemental information for this article can be found online at http://dx.doi.org/10.7717/peerj.18381#supplemental-information.

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
