# Peer review of "Effect of vitamin C injections on exercise muscular performance and biochemical parameters in Trichinella spiralis-infected mice"

_PeerJ, doi:10.7717/peerj.18381_

## Round 0.1 · original submission · Major Revisions

Please address concerns of all reviewers and amend manuscript accordingly.

**Language Note:** The review process has identified that the English language must be improved. PeerJ can provide language editing services - please contact us at [email protected] for pricing (be sure to provide your manuscript number and title). Alternatively, you should make your own arrangements to improve the language quality and provide details in your response letter. – PeerJ Staff

Reviewer 1 ·

Basic reporting

The aim of this manuscript was to evaluate the effect of intraperitoneally administered vitamin C on the course of muscular phase of trichinellosis in mice experimentally infected with T. spiralis.
To obtain the results authors performed biological assay in which mice had been experimentally infected with T. spiralis and then (after fifteen days) they received 10mg/ kg b.w. of vitamin C in daily doses. Mice which have been infected and recived saline and uninfected animals served as control groups. To assess muscle damage and muscle larval burden, several techniques such as: compression method (for larvae counting), standard histopathological analysis, simple biochemical tests (for the determination of creatine kinase, aspartate aminotransferase, malondialdehyde and superoxide dismutase levels) as well as so-called muscle performance tests were introduced.
Finally, authors achieved interesting results and showed that, in T. spiralis-infeced mice, daily intraperitoneal administration of vitamin C may reduce burden of larvae in muscles, improve muscle strength or reduce muscle damage.
In fact, effective treatment of trichinellosis, especially in its early (intestinal) phase, is still a challenge. Due to the strong tissue (muscle) damaging effects during Trichinella infection, symptom relief is also an important matter. In this context the aim of the present work is reasonable.
Notwithstanding the above, this reviewer (TR) has a few comments, suggestions and questions about the manuscript, and all of them should be treated by the authors as major points.
1. First and the most important thing is: Why did the authors use the compression method instead of standard digestion assay in order to assess the number of larvae in the muscles? To assess the larvae number, 0.1 gram of tissue is not enough and rather does not give reliable results.
2. Authors wrote that they were able to count encysted and newborn larvae (table 2 and figure 4). I think that authors wrongly interpret the newborn larvae (NBL) phase of the infection and NBL as the developmental stage of the parasite. Newborn larvae are released from the female (enteral phase of infection), and then as migratory larvae they are present in the blood and lymphatic system. When they invade the muscle fibers, in fact, they became muscle larvae (ML) with preinfective stage (they become infective after 15-17 days after entering muscle cell). Probably, while observing the muscle tissue under the microscope, the authors saw muscle larvae that had encysted and those that were not encapsulated because they were born later. Generally speaking, the female is able to give birth to larvae for 1-2 months after infection (depending on Trichinella species and species of the host), although her fertility is the highest in the first two weeks. In fact, by using standard light microscope and compression technique, similar view (non-encapusalted larvae) would apply to T. pseudospiralis larvae as well. In order to assess the number of migratory larvae or newborn larvae, authors should examine the lymph and blood, possibly the intestines as well which are both quite a difficult task. If the authors would like to assess the muscle larvae reduction rate under the influence of vitamin C, I strongly recommend using a standard artificial digestion technique by digesting, for instance, the entire thigh, or entire thigh and entire diaphragm.
3. It is not exactly clear how the authors assessed the larvae as: (1) intact, (2) deformed and (3) vacuolated using simple histopathology technique (figure 6). Please add more information about that issue. Moreover, were deformed larvae capable of infection?
4. Why did the authors choose 10 mg/kg of body weight dose of vitamin C?
5. Abstract: change ‘briefly’ to ‘summing up’
6. Introduction: what does ‘Trichinella-related nematodes’ mean here? Change to (for instance): ‘Nematodes of the genus Trichinella….’
7. Line 45: All Trichinella species are pathogenic to humans.
8. Line 50-51: Authors wrote: ‘In the skeletal muscles, the newborn larva develops into mature cells and modifies the muscle cells into….’ What do ‘mature cells’ mean here? It should be: ‘In the skeletal muscles, the Trichinella larvae develop into fully infective muscle larvae and modifies the muscle cells into…’
9.Line 63: change ‘our bodies’ to human body’
10. Line 97: change ‘…their daily injections for two weeks’ to ‘their daily injections (vit. C or saline) for two weeks’
11. Statistical analysis:
a. How the distribution and homogeneity of variances were checked?
b. Please give standard deviations instead of standard error of the mean.
c. Did the authors compare two different time points within the same group (mice infected with T. spiralis and treated with vitamin C)?
12. Fig 6: title and footnote contain the same information.
13. All abbreviations should be expanded if they appear for the first time in the manuscript.

Experimental design

Section 'basic reporting' includes my entire review report.

Validity of the findings

Section 'basic reporting' includes my entire review report.

Additional comments

Section 'basic reporting' includes my entire review report.

Reviewer 2 ·

Basic reporting

The authors present an interesting topic on the effect of vitamin C supplement on Trichinella spiralis infected mice and evaluated its efficacy by exercise muscular performance, histopathology and some biochemical parameters.
However, the following points have to be considered.
Revise the whole manuscript for English language mistake.
The ethical approval is obtained from Suez Canal University while the authors affiliation and the work was done in Port-Said University (line 85).

Experimental design

Materials and methods are lacking many important details:
- Duration and time of vitamin C supplement (when and for how long??)
- Regarding hanging and weights tests: which muscles are examined by each test? Time of test performance before and after infection).
- Add figures for the mice while performing these tests.
- Electromyography is required for different muscles; triceps, gastrocnemius, masseter, and ribcage
- Lines 94-97 are repeated.

Validity of the findings

Rewrite the results in a better way. Avoid starting your sentence with; fig … shows or according to fig…..
The authors described the larvae ad newborn and encysted. This is a big mistake. Newborn larvae are produced by Trichinella adult females during the intestinal phase. The specimens of this work is obtained from the muscles 30 days post infection. What you describe as newborn larvae are those released from the capsule after digestion of muscles. So, delete the word ‘newborn’ from the whole manuscript and use encysted larvae instead and apply this on the results and correct them accordingly. Use the total number of larvae for assessment of reduction in the count.
Body weight is mentioned in the results and not described previously in the methods as a parameter for evaluation.
Mention clearly, the parameters that you chose for the histopathological classification lines 145. Some of them are shown in the figures.
Line 164: compare these results with the negative control mice.

Additional comments

In the discussion section, you didn’t explain how vitamin C eliminate Trichinella muscle larvae after their settlement if the muscles. It is accepted that Vitamin C reduce the inflammatory infiltration but reduction in the number of larvae is not justified especially that the authors didn’t investigate the intestinal phase.
Parasite names are written in italic
Line 37: ‘in conclusion’ instead of ‘briefly’
Line 38: ‘improving’ instead of 'enhancing’
Line 43: Change ‘Trichinella-related nematodes’ …. to ‘Trichinella is among …….
Line 126: ‘post infection’ instead of ’ infection’
Line 183: delete ‘ each type of larvae’ and delete ‘ three forms’ as I mentioned before.
Line 431: spelling mistake
Update the references. Too much old references 1989, 1996, 1997….
Figures:
Fig 2: mean body weight. It is better to present these findings in a different way to show the changes in each group at each time point.
Change the title of fig 4 according to what I mentioned above. Both larvae shown in the figures are muscle larvae.
Delete fig 5
Figure 6: correct ‘different encysted larvae types’. Delete the word ‘developing’ .
Fig 6D: shows percentages or numbers of larvae which is not described previously in the results section and what are the characters used to describe them. They are not mentioned.
Similarly, fig 7C: these changes are not mentioned in the methods or results.
Fig 8: # significant difference from…. Complete the sentence.
Poor quality of histopathology figures
Table 1: add the number of mice in each group
Table 2: use ‘ mean larval count’ instead of ‘larvae average’
As mentioned before : use the total number for comparison

·

Basic reporting

More specific impormation need to be added as per the objective of the dstudy.

Experimental design

experimental designe seems to be fine.

Validity of the findings

through statistical analysis need to be explained in the discussion

Additional comments

Revised Title:
"Effect of Vitamin C Injections on Exercise Muscular Performance and Biochemical Parameters in Trichinella spiralis-Infected Mice". There is need for language improvement throughout the manuscript.
Abstract:
• Ensure results in the abstract are quantified and specific.
Language Improvements:
• Line 24: "vitamin C's potential" -> "potential of vitamin C"
• Line 25: "mice's muscles" -> "muscles of mice"
• Line 27-28: "Two weeks post-infection, each group was intraperitoneally injected daily for two weeks." - Clarify what was injected (e.g., "with vitamin C injections").
• Line 46: Specify as "Trichinella spiralis" the first time it is mentioned.
• Line 49: Similarly, use "Trichinella spiralis" if the sentence starts with the genus.
• Line 72: "vitamin C's ability" -> "ability of vitamin C"
• Line 100: "group" -> "groups"
• Lines 102-103: "Fifteen days post-infection, all the mice began to receive their daily injections for two weeks." - Specify what injections (e.g., "received daily vitamin C injections for two weeks").
• Line 167: "According to Figure 2" -> "According to the results"
• Line 239: Clarify whether it refers to "efficiency of vitamins" or "immunomodulatory effect".
• Lines 327-331: Ensure that all references are cited appropriately in the text. Reference “Binfaré RW, Rosa AO, Lobato KR, Santos AR, Rodrigues ALS. 2009. Ascorbic acid administration produces an antidepressantlike effect: evidence for the involvement of monoaminergic neurotransmission. Progress in Neuro-Psychopharmacology and
Biological Psychiatry33(3):530-540.” Is not cited in text
Email Correspondence:
• Consider reducing the number of emails sent to the first author to one or two, and consolidate communication for the corresponding author if possible.
By implementing these changes, the manuscript should become clearer, more specific, and address the reviewer's suggestions effectively.

---

## Round 0.2 · Major Revisions

Please address remaining concerns of the reviewer and amend manuscript accordingly. Please note that it is not necessary to remove the part of larval count as recommended by the reviewer unless you think that this is a reasonable request.

Reviewer 2 ·

Basic reporting

It will be better if the authors remove the part of larval count and stay confined to the changes in the exercise muscular performance and biochemical parameters according to the title.

Experimental design

How did the authors use Vitamin C effervescent tablets (prepared for oral use) as intraperitoneal injection? and justify the use of intraperitoneal injection.

Validity of the findings

The authors mentioned that [the larvae were categorized statistically based on the degree of integrity of their internal structures. They were divided into three groups: fully intact larvae, partially intact or deformed larvae, and fully degenerated ones inside the capsules.]
Kindly mention the criteria of each one of them. In other words, when did you consider the larvae deformed of fully degenerated?

Line 139: delete newborn larvae. No newborn larvae could be detected in your current study.

How can vitamin C eliminate Trichinella muscle larvae after their settlement in the muscles?
The authors showed massive reduction in the larvae which is not explained. Can muscle larvae vanish after settlement in the muscles!!!!

Additional comments

no comment

·

Basic reporting

all comments and suggestions have been addresssed properly by the authors.

Experimental design

Ok

Validity of the findings

fine

Additional comments

Authors have throughly addressed the comments/suggestions. All the possible corrections have been made. Manuscript has been improved a lot.

---

## Round 0.3 · Major Revisions

As you can see, the reviewer still has serious concerns that must be addressed.

Reviewer 2 ·

Basic reporting

Thanks for the correction done by the authors.
However, the massive reduction in the muscle larvae after settlement in the muscles in vit C treated group is still not explained.
The treatment started fifteen days post-infection, the larvae were already left the intestine and reached the muscles. No effect is expected at this stage regarding the muscle larval count.

Moreover, the reference cited by the authors “Influence of vitamin C on the resistance of rats to Trichinella spiralis infection, 1986, J J Senutaité, S Biziulevicius” is not available and even its abstract is not available. So, you can’t use it to support your findings.

I suggested to remove ‘the part of larval count’ to overcome this conflicting point.
You can keep the part describing the degeneration that appeared in these larvae.

Experimental design

Corrections done

Validity of the findings

The effect of vit c on the number muscle larvae is not valid.

Additional comments

none

---

## Round 0.4 · accepted · Accept

In my view, remaining issues were adequately addressed, and the revised manuscript is acceptable now.